# Reproductive Behaviours and Potentially Associated Sounds of the Mottled Grouper *Mycteroperca rubra*: Implications for Conservation

**Elena Desiderà** [1,2,3,*], **Carlotta Mazzoldi** [2,4], **Augusto Navone** [5], **Pieraugusto Panzalis** [5], **Cédric Gervaise** [6], **Paolo Guidetti** [1,3,7,*] **and Lucia Di Iorio** [6,8]

1. Department of Integrative Marine Ecology (EMI), Stazione Zoologica Anton Dohrn–National Institute of Marine Biology, Ecology and Biotechnology, Genoa Marine Centre, Villa del Principe, Via San Benedetto 2, 16126 Genoa, Italy
2. Department of Biology, University of Padova, Via U. Bassi 58/B, 35121 Padova, Italy; carlotta.mazzoldi@unipd.it
3. UMR 7035 ECOSEAS, CNRS, Université Côte d'Azur, Parc Valrose 28, Avenue Valrose, 06108 Nice, France
4. CoNISMa, Interuniversity National Consortium for Marine Sciences, Piazzale Flaminio 9, 00196 Rome, Italy
5. Marine Protected Area of Tavolara-Punta Coda Cavallo, Via Dante 1, 07026 Olbia, Italy; direzione@amptavolara.it (A.N.); ambiente@amptavolara.it (P.P.)
6. CHORUS Institute, 5 Rue Gallice, 38016 Grenoble, France; cedric.gervaise@chorusacoustics.com (C.G.); lucia.diiorio@chorusacoustics.com (L.D.I.)
7. National Research Council, Institute for the Study of Anthropic Impact and Sustainability in the Marine Environment (CNR-IAS), Via De Marini 16, 16149 Genoa, Italy
8. UMR 5110 CEFREM, CNRS, Université de Perpignan, 52 Avenue Paul Alduy, 66860 Perpignan, France
* Correspondence: elena.desidera@szn.it (E.D.); paolo.guidetti@szn.it (P.G.)

**Abstract:** Most grouper species worldwide are threatened by overfishing. Effective marine protected areas (MPAs) are known to enable population recovery, and consideration of vulnerable species' reproductive behaviours is fundamental to monitoring and management plans. Many groupers produce sounds associated with reproductive behaviours. Recording these sounds helps to locate spawning sites and improve management efforts to ensure reproduction and viability. This study focuses on a poorly studied yet likely vulnerable grouper species, *Mycteroperca rubra*, providing novel insights into its reproductive biology by combining underwater visual census surveys, direct visual observations and passive acoustic monitoring within a Mediterranean MPA during two consecutive summers (2017 and 2018). Results indicate that *M. rubra* individuals were more abundant and larger at one of the protected study sites, where they also occasionally formed unusual aggregations (<30 individuals), likely for spawning. These aggregations and the observation of courtship behaviours suggest that *M. rubra* spawns in the surroundings of this study site. Moreover, grouper-like unknown sounds were recorded exclusively at this site, suggesting they are associated with *M. rubra* courtship behaviours. Therefore, this study provides a basis for validating *M. rubra* sound production and supports the monitoring of its spawning sites via passive acoustics to improve MPA conservation effectiveness.

**Keywords:** Epinephelidae; passive acoustic monitoring; spawning aggregations; marine protected areas; Mediterranean Sea; courtship behaviour; fish sounds

## 1. Introduction

The marine realm is facing multiple threats posed by increasing cumulative human-related impacts, primarily over-exploitation, habitat degradation and climate change [1–3]. Marine protected areas (MPAs) are conservation and management tools aimed at protecting biodiversity, promoting healthy and resilient marine ecosystems, and providing societal benefits [4]. If properly designed, well-managed and effectively enforced, MPAs have

proven to enhance the abundance, size and biomass of ecologically and economically valuable fish species, such as high-level fish predators [5–7].

The long-term viability of exploited fish populations depends on maintaining viable breeding densities; therefore, knowing where and when these fish reproduce is fundamental for guiding conservation/management actions. The total prohibition of fishing and other extractive activities (i.e., full protection) at key reproductive sites may help rebuild locally depleted or declining populations through increased reproductive output within the MPA and subsequent enhancement in recruitment both inside and outside the MPA [8–12]. When designing new MPAs and/or enforcing/expanding already established MPAs, the inclusion of spawning grounds within fully protected areas is strategic to dramatically increase MPA ecological benefits [13] and to meaningfully support the achievement of international conservation targets. As an example, the recent EU Biodiversity Strategy for 2030 has set the goal of extending protection to at least 30% of European seas, of which one third should be fully protected, by 2030 [14,15].

Recent evidence suggests that management frameworks should account for fish reproductive behaviours alongside other more traditional life history traits in order to be effective [16]. Many marine fish species, such as groupers (Epinephelidae), form spawning aggregations, making them particularly vulnerable to overfishing [17,18]. To be defined as such, a spawning aggregation must: (i) be a repeated and predictable concentration of conspecifics, gathered for the purpose of spawning; (ii) be characterised by, at least, a three-fold greater individual abundance than at non-aggregation times; and (iii) result in the release and subsequent dispersal of large numbers of offspring [19,20]. Based on the distance travelled by individuals to spawning sites, their duration and frequency of occurrence, spawning aggregations range from resident to transient aggregations, with some falling in between these two main categories [16,19]. Large-sized fish species, including groupers, typically travel long distances (tens to hundreds of kilometres) to form transient aggregations that last from days to weeks during one to a few months each year [19]. Transient aggregating fish species show the greatest vulnerability to fishing and are thus generally more likely to be overfished than those forming resident or mixed aggregations [16]. Regardless of the type of aggregation, several fish species are known to aggregate at the same spawning sites, either sequentially or simultaneously [17]. As breeding hotspots, multi-species spawning sites are acknowledged as focal points for conservation, where localised protection efforts can disproportionately benefit vulnerable populations [9,21,22] and biodiversity conservation [8].

Many marine fish are known to produce sounds when displaying reproduction-related behaviours, such as courtship activities [23–31]. Passive acoustic monitoring (PAM) has thus emerged as a valuable non-invasive tool in fishery and conservation science for the identification and protection of spawning sites of exploited soniferous fish and the monitoring of their spawning dynamics across time and space [30–32]. PAM is particularly useful when soniferous fish exhibit high site fidelity and aggregate to spawn at predictable times and places [31]. This is the case with many grouper species [22]. To date, sound production associated with courtship behaviours has been widely documented for several groupers dwelling in tropical and subtropical areas [33–39] and PAM was found extremely useful in discriminating the acoustic and reproductive activity of different grouper species [28,40,41]. Grouper courtship behaviours encompass a variety of elaborate interactions between males and females, often exhibiting distinctive colour patterns (sexual dichromatism), that are performed by males to attract females and encourage mating activity [39,42–44]. The grouper courtship behaviours that have been linked to concurrent sound production are commonly characterised by lateral displays, body quivering or twitching, head shakes and burst rises, thus consisting in a combination of visual and acoustic stimuli [33–39].

In the Mediterranean Sea, information on sound production by groupers remains scant. Of the six native Mediterranean grouper species [45], only the dusky grouper, *Epinephelus marginatus*, is known to emit sounds [46]. Particularly, one of the two types of sounds that have been attributed to the dusky grouper is associated with courtship

activity; courting males of *E. marginatus* approach females with a lateral display and then shake the rear part of the body ("ritualised caudal flapping") [46–48]. *E. marginatus* is generally the most common and thus the most studied species in the Mediterranean [49]. However, depending on the Mediterranean sub-areas/sites, other grouper species can be more abundant than or as common as the dusky grouper, such as the mottled grouper, *Mycteroperca rubra* (Bloch, 1793).

*M. rubra* is a sub-tropical grouper native to the Mediterranean Sea and, being thermophilic, it is more common in the southern than in the northern Mediterranean [50–53]. However, due to ocean warming, it is expanding its range northward with new records in the northwestern Mediterranean made over the past two decades [53–56]. Like most groupers [22], the mottled grouper is a protogynous hermaphrodite, changing sex from female to male [57]. Females are reported to attain sexual maturity around 35 cm TL and change sex around 40–65 cm TL [52,57].

Given its maximum total length (~80 cm, [52]), *M. rubra* is a fishing target where it commonly occurs [53,58–60]. In the Mediterranean Sea, it is reported to have been targeted by spearfishers for ~50 years [56]. At the global level, the lack of fisheries data does not make it possible to draw any conclusion concerning the current population trend. This is why, to date, fishing is not considered a major threat for *M. rubra*, being classified as of Least Concern in the IUCN Red List [53]. However, there is a risk that localised population declines may have occurred and/or are occurring in areas with heavy fishing pressure [53], similarly to *E. marginatus* [5,61].

Effective protection from fishing in northern Mediterranean MPAs and seawater temperature increase may have contributed to the northward range expansion of *M. rubra* [56]. In fish monitoring studies, individuals of *M. rubra* were more abundant and larger in size within MPAs than in their fished surroundings [7,55,62–64]. *M. rubra* is more commonly observed as a solitary fish, in isolated pairs or small groups (<10 individuals) [52], even in MPAs [55,63]. Larger gatherings of 10–100 s of individuals have been documented in only four MPAs, one located in the eastern Mediterranean (off the Israeli coast [52]) and three in the western Mediterranean Sea (Scandola Nature Reserve [55]; Portofino MPA and Secche di Tor Paterno [56]). Additional aggregations were observed in unprotected areas where the species is more common, that is, off Mediterranean Turkey [53] and the Israeli coast, although in the latter case they were less regularly observed and generally less dense compared to those found in the MPA [52]. As in the case of other grouper species, these gatherings might represent spawning aggregations [17] because their time of occurrence corresponds to the spawning season of the species (spring in the southern and probably summer in the northern Mediterranean [57]). Moreover, individuals within such aggregations displayed courtship behaviours along with changes in body colour pattern [52,55]. In fact, the basic colour pattern of adults is mottled with pale spots, but two other colour patterns have been observed during the reproductive season: a dark uniform pattern and a silver pattern [45,52,55]. Adults of *M. rubra* displaying the silver pattern have been observed quivering while swimming very quickly and rising in the water column, suggesting that this peculiar display is a courtship behaviour [55]. The presumptive spawning aggregations observed off Israel lasted from a few days to two weeks during five–six months [52], suggesting that *M. rubra* is a transient aggregating species [16,19].

To date, only one study conducted in 1999–2002 has provided information on the reproductive biology of *M. rubra*, also documenting the occurrence of its largest (up to 500 individuals) putative spawning aggregations [52]. No data on the mating patterns of *M. rubra* are currently available. Therefore, knowledge of the spawning behaviours and dynamics of *M. rubra* is currently scarce. However, such knowledge is crucial to assess the vulnerability to fishing of poorly studied fish species and identify areas where targeted protection of spawning fish may be needed [16].

In a preliminary study aimed at locating grouper aggregations occurring within the MPA of Tavolara–Punta Coda Cavallo (Italy, NW Mediterranean Sea), three locations were selected as potential grouper aggregation sites based on local knowledge [65,66] and

the available literature [67]. By combining underwater visual census, direct behavioural observations and PAM at the three selected sites, this study aims at (i) collecting occurrence data of *M. rubra* indicating the formation of potential spawning aggregations, (ii) describing reproduction-related behaviours and (iii) providing evidence of potential courtship-associated acoustic activity.

## 2. Materials and Methods

### 2.1. Study Sites

The present research was conducted within the Marine Protected Area of Tavolara–Punta Coda Cavallo (40°53′ N, 09°41′ E; hereafter TPCCMPA), located in the western Mediterranean Sea (Italy). The TPCCMPA was established in 1997 but became effectively managed and enforced around 2003-2004. In accordance with Italian law, it includes three types of zones (A, B and C) with a decreasing protection gradient from A to C zones (Figure 1, see details in [65]). Spearfishing as well as fishing groupers by any means are prohibited throughout the TPCCMPA.

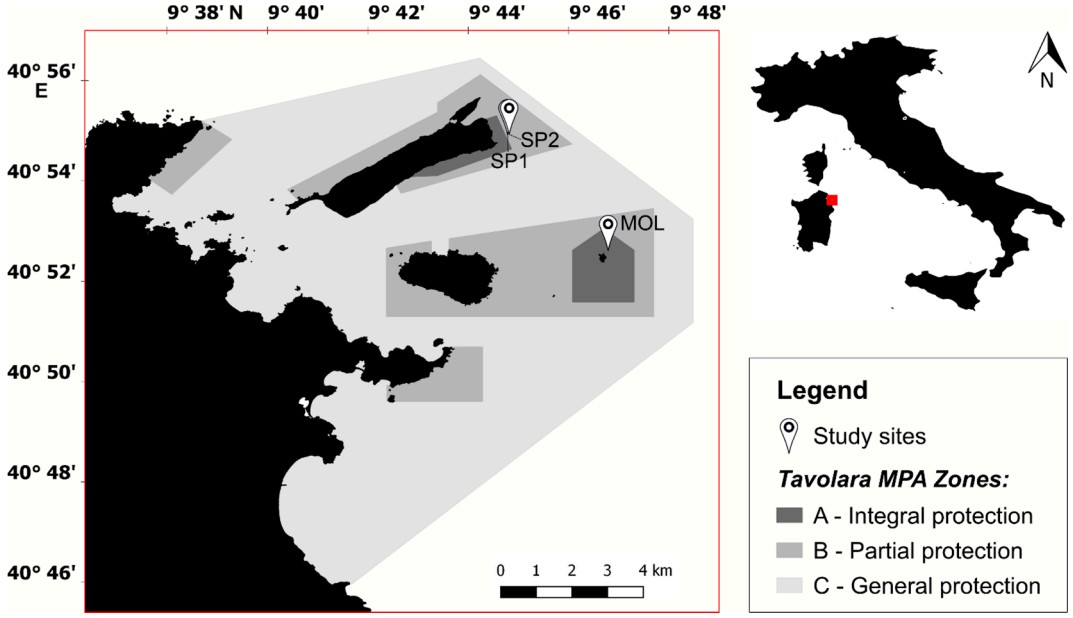

**Figure 1.** Zonation of Tavolara–Punta Coda Cavallo Marine Protected Area (TPCCMPA). Study sites are indicated: SP1 = Secca Papa 1; SP2 = Secca Papa 2; MOL = Molarotto. Zone A: fully protected (no-take and no-access) zone; Zones B and C: buffer zones where human activities are restricted. In the B zone, only licensed local small-scale fishing and diving are allowed, while in the C zone, regulated fishing is also open to recreational fishing.

This research was conducted during the summers of 2017 and 2018 at three protected sites, identified based on regular sightings of many large individuals, likely spawners, of different grouper species [67,68]. The study sites were (Figure 1): (1) "Secca Papa 1" (SP1, B Zone); (2) "Secca Papa 2" (SP2, B Zone), consisting of two rocky banks where diving is allowed, while all fishing is forbidden within a radius of 100 m; and (3) "Molarotto" (MOL, A Zone), consisting of a specific group of fully protected rocky outcrops named after the nearby granitic islet. The study sites are characterised by rocky substrates that differ in terms of lithological composition (limestone at SP1 and SP2 versus granite at MOL), depth range (15–45 m at SP1 and SP2 versus 20–30 m at MOL) and topographic features (for more details see [65]). The rocky banks of SP1 and SP2 were treated as separate study sites, despite being adjacent, since indications of divergent habitat preferences (spatial segregation) between multiple aggregating fish species have been collected therein (Desiderà, unpublished data) and because the local bottom morphology (underwater pinnacles) strongly influence sound propagation and signal detection.

### 2.2. Underwater Visual Census (UVC)

Overall, three trained SCUBA divers conducted UVC surveys to estimate the average density and size-frequency distribution of *M. rubra* at the three study sites (sampling dates in Supplementary Materials Figure S1a). Specifically, at each site and during each dive, four replicates of a standard $25 \times 5$ m$^2$ strip transect (surface area 125 m$^2$) [69] were randomly performed by a single diver, avoiding surveying adjacent or overlapping surfaces to guarantee replicate independence. Replicates were conducted on homogenous rocky bottoms by swimming along the rocky reef profile at a 15–35 m depth range. The number and size of the strip transects were set based on the relatively restricted area to be surveyed and the constrained diving time at the sampled depth range. Within each transect, the number and size of *M. rubra* individuals were recorded. Fish length (total length, TL) was estimated and recorded using 5 cm size classes [68]. It is highly unlikely that the same individuals were counted multiple times, since the study species is less territorial and more closely associated with the water column compared to other groupers, such as *E. marginatus* [51]. In order to avoid temporal autocorrelation, the same study site was re-sampled after at least two days. All UVC dives were conducted between 08:00 and 13:00 Central European Summer Time (CEST, i.e., UTC + 2 h). Hereafter, all times are referenced to CEST.

Differences in *M. rubra* density across the three sampling sites, for both years combined, were tested by a Kruskal–Wallis test, as data were not normally distributed, using R Version 4.1.2 [70]. The Kruskal–Wallis test was followed by a Dunn *post hoc* test, as it is appropriate for groups with unequal numbers of observations, and $p < 0.05$ was considered significant.

### 2.3. Observations of Reproductive Behaviours

In addition to UVC surveys, direct underwater observations on mottled grouper behaviours were conducted at the study sites within three main time slots: after sunrise (from 6:00 to 9:00), during daytime (from 9:01 to 13:00) and before sunset (from 17:00 to 21:00) (Supplementary Materials Figure S1b). Three trained SCUBA divers (two observers per dive) collected behavioural data while exploring each site without specific restrictions of depth or time. Observational notes were recorded onto an underwater plastic board and included: (i) the number of individuals aggregating; (ii) observations of individuals changing colour patterns (mottled, dark and silver) and/or just displaying the silver pattern; and (iii) the occurrence of courtship activity, which consisted of fish with the silver pattern quivering and rising in the water column. These data were used as indirect evidence of reproductive activity. Presumed spawning aggregations and courtship behaviours of *M. rubra* were also recorded using a high-definition camera (Sony Cyber-shot DSC-RX100). Videos were analysed to support/complement the observational notes. The timing of observed courtship behaviours and/or aggregations were noted on a spreadsheet for comparison with the acoustic recordings. Particular attention was given to the body-shaking behaviour because it has been associated with the contraction of the sonic muscles responsible for sound production of known vocal groupers [33–39,71,72].

Since seawater temperature is known to affect muscle contraction rate and thus sound characteristics [73], during all dives, bottom seawater temperature data were retrieved from the dive computer (Suunto Vyper). In 2018, water temperatures were also recorded with a $\pm 0.5$ °C resolution every hour by a sensor (iButton® device, type DS1922L, Maxim Integrated Products, Inc, San Jose, CA, USA), deployed at MOL at 29.6 m of depth on 1 August and recovered on 10 September.

### 2.4. Acoustic Recordings

#### 2.4.1. Data Collection

Passive acoustic recordings were collected at the study sites. Recorders were all moored between 25 and 40 m depth using sandbags. The recording equipment consisted of four autonomous underwater acoustic recorders that were deployed using a rotating schedule (available in the Supplementary Materials Figure S2). A SongMeter SM2M (Wildlife Acoustics

Inc., Maynard, MA, USA) and three EA-SDA 14 recorders (RTSYS®, France) were used. The SM2M recorder was equipped with a wideband omnidirectional hydrophone HTI-96-MIN (High Tech Inc., Long Beach, CA, USA; receiver sensitivity: −163.4 dB re. 1 µPa/V, flat frequency response: 2 Hz-0 kHz, 16-bit resolution). It was programmed to record with a 96 kHz sampling frequency and a 16-bit resolution. The EA-SDA 14 recorders were equipped with an HTI-92-WB hydrophone (High Tech Inc.; receiver sensitivity: -55 dB re. 1 µPa/V, flat frequency response: 5 Hz-0 kHz). They recorded at a 24-bit resolution and a 78 kHz sampling rate. At MOL, one of the EA-SDA recorders was connected to a battery pack that allowed almost continuous recording in August 2017. Recording duty cycles differed among sites depending on the recording devices used and ranged from 10 min every 3 min to 10 min every 30 min (Supplementary Materials Figure S2).

### 2.4.2. Data Analysis

Since most fish vocalise and hear in the low frequency range (<2000 Hz [23,74]) and because grouper calls are low in frequency (<200 Hz) [46,75], audio recordings were down-sampled to 4 kHz. Using the software Raven PRO 1.5 (The Cornell Lab of Ornithology, Ithaca, NY, USA), spectrograms (Hamming window, fast-Fourier-transform (FFT) length = 256 samples/points) and associated oscillograms were visually inspected to (i) identify the known dusky grouper calls [46,75] and sounds sharing typical acoustic characteristics with grouper calls but attributable to other grouper species [35–37,39,40,46]; and (ii) characterise these grouper-like sounds (differing from dusky grouper calls) in terms of spectral and temporal features. Specifically, the sound duration (i.e., time span from the first to the last pulse peak; ms), peak frequency (Hz), frequency 5% (Hz), frequency 95% (Hz), bandwidth 90% (Hz) and, in pulse trains, the number of pulses and the inter-pulse interval (IPI, the peak-to-peak interval; ms) were measured. A MATLAB routine and interface was then used to accelerate the audio-visual inspection of the acoustic recordings. This routine converted each audio file into a sequence of 10 s images, each of which consisted of (i) a spectrogram reporting the received sound pressure levels (RLs) as a function of time and frequency (FFT size 256, Kaiser window with 80% overlap), (ii) the same RL spectrogram, in which the background noise was removed to better visualize individual calls (see the method described in [76]) and (iii) three one-second RL spectrograms of the sounds with highest signal-to-noise-ratio within the 10 s frame (Supplementary Materials Figure S3). This MATLAB interface allowed us to select the sound types detected in each 10 s spectrogram. Sound type selections were then summarized in csv output files used for conducting analyses in R 4.1.2 [70].

### 2.4.3. Temporal Patterns of Sound Production

The first field observation of a potential transient spawning aggregation [19,20] of *M. rubra* in the TPCCMPA was made in July 2017. After having preliminarily inspected and analysed the full day audio files recorded at MOL on 7 July (~11 h of recordings), the acoustic data collected over the same month were analysed to characterise temporal patterns of crepuscular and nocturnal sound production at the three study sites. In fact, grouper species emit sounds mainly around sunset and at night [39,46,77], when sound production is also less masked by anthropogenic noise (i.e., nautical activities) than during the day. Therefore, only recordings obtained from 18:00 until 8:00 were processed. Moreover, since recording cycles differed across sites, the number of potential *M. rubra* sounds per effective minute of recording (weighted sound abundance) was computed by dividing the cumulative number of sounds per sound type by the true duration of acoustic measurements. Time series of weighted sound abundances were generated to visualise patterns in sound production.

### 2.4.4. Relationship between Visual Observations and Presumed Acoustic Behaviour

To investigate the potential sound production activity associated with courtship-related behaviours (aggregations, courtship displays and colour changes), dives for which

concomitant acoustic recordings were available were used for the analyses. Since diving efforts were greater in 2018 than in 2017, only the dives and concurrent recordings performed in July-August 2018 were included. Based on the presence or absence of courtship behaviours, dives were labelled as C ("Courtship") or NC ("No-Courtship") on the underwater board, respectively. Out of all the audio files, only those recorded within a three-hour interval overlapping each dive were considered. Audio recordings were labelled as C/NC according to the dive to which they corresponded. For each three-hour interval, the potential *M. rubra* sounds were selected using the MATLAB interface mentioned earlier and weighted sound abundances were then computed.

## 3. Results

### 3.1. Underwater Visual Census (UVC)

A total of 88 UVC transects were conducted at MOL, 64 at SP1 and 52 at SP2 (Table 1).

**Table 1.** Number of UVC transects conducted to record the mottled grouper size and density in the summers of 2017 and 2018 at the three study sites.

|  | **Site** | **2017** | **2018** | **Total** |
|---|---|---|---|---|
| Total N. UVC transects | MOL | 32 | 56 | 88 |
|  | SP1 | 24 | 40 | 64 |
|  | SP2 | 24 | 28 | 52 |
| N. UVC transects where *M. rubra* was recorded | MOL | 5 | 18 | 23 |
|  | SP1 | 0 | 5 | 5 |
|  | SP2 | 0 | 3 | 3 |

Overall, the mottled grouper was censused in 31 transects out of the 204 performed at the three study sites. The sightings of *M. rubra* individuals were more common at MOL than at SP1 and SP2. At least one mottled grouper was recorded in 23 transects at MOL, 5 at SP1 and 3 at SP2 (Table 1).

In summer 2017, the mottled grouper was censused only at MOL with an average density of $0.25 \pm 0.12$ (mean $\pm$ SE) individuals per 125 m$^{-2}$ (transect surface area), while in summer 2018, the recorded grouper average density was $0.63 \pm 0.15$ at MOL, $0.25 \pm 0.12$ at SP1 and $0.11 \pm 0.06$ at SP2 (Figure 2).

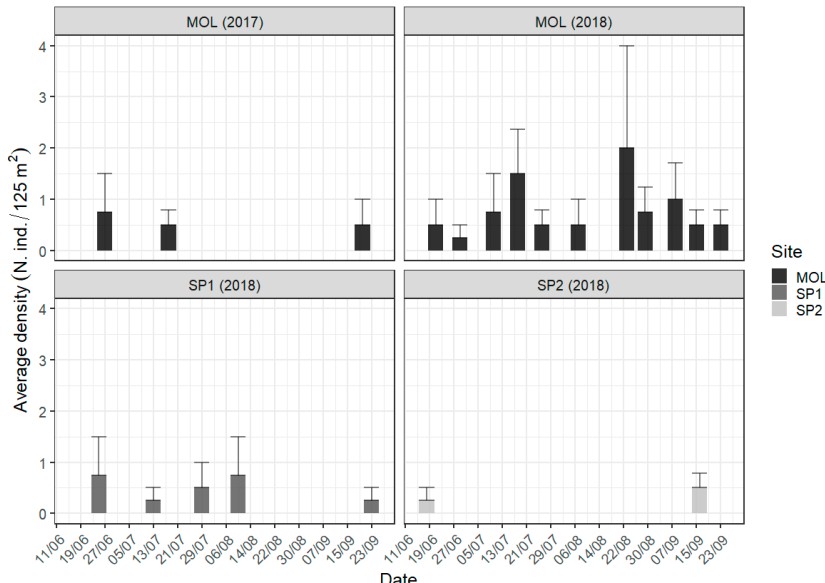

**Figure 2.** Average densities of *M. rubra* ($\pm$SE) recorded at MOL during the summer of 2017 and 2018, and at SP1 and SP2 in the summer of 2018. No mottled grouper was censused at SP1 and SP2 in the summer of 2017. Four replicates (strip transects) per study site per sampling date were conducted.

Statistically significant differences in grouper density were found across the study sites for both years combined (Kruskal–Wallis chi-squared = 14.41, df = 2, *p* = 0.0007). The Dunn test found a significant difference only between MOL and the other two study sites (Figure 3, Supplementary Materials Table S1).

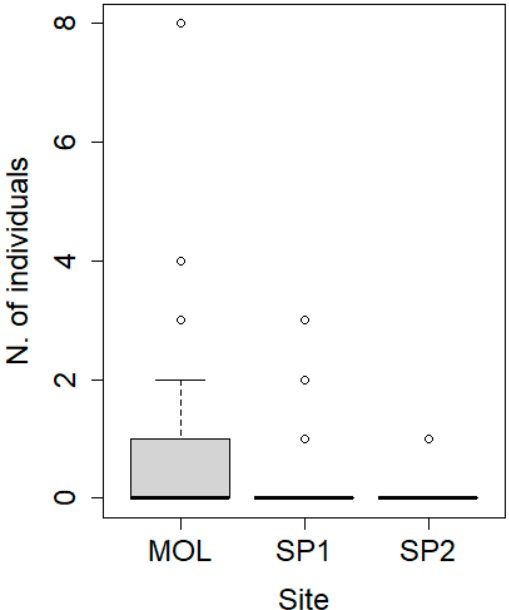

**Figure 3.** Box-plot of the cumulative abundance of individuals of the mottled grouper censused per study site considering both years (2017 and 2018). Number of replicates (strip transects) per site: 88 at MOL, 64 at SP1 and 52 at SP2. Mottled grouper density was significantly different at MOL compared to the other sites (Kruskal–Wallis chi-squared = 14.41, df = 2, *p* = 0.0007).

In summer 2018, the recorded size distributions of *M. rubra* differed among the three study sites, with the largest individuals recorded at MOL. Individual total lengths (TL) ranged from 45 to 85 cm at MOL and from 55 to 65 cm at SP1 and SP2 (Figure 4).

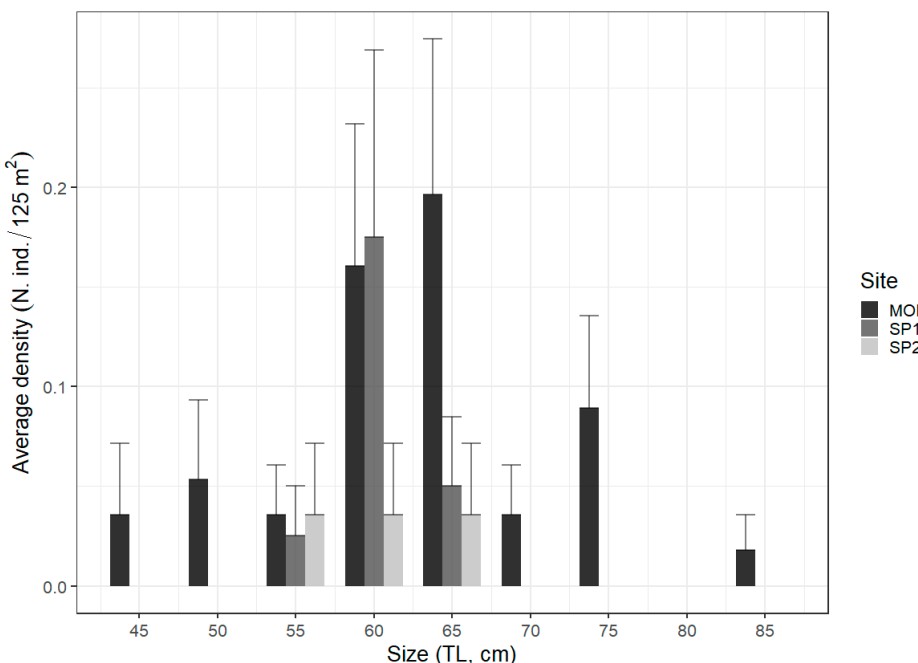

**Figure 4.** Mottled grouper average densities (±SE) recorded per strip transect (surface area 125 m$^2$) and size class (45–85 cm TL) at the three study sites in summer 2018.

### 3.2. Behavioural Observations

Direct observations were conducted during 120 dives (*n* = 40 in 2017, *n* = 80 in 2018), sub-divided into the three time slots as follows: 32 dives after sunrise, 27 dives during daytime and 61 dives before sunset (Table 2). Overall, 42 dives were performed at MOL, 46 at SP1 and 32 at SP2 (Table 2). The average dive duration was 36:12 ± 6:17 (mean ± SD) minutes.

**Table 2.** Number of dives conducted during three time slots to directly observe the mottled grouper behaviours in the summers of 2017 and 2018 at the three study sites. Time slots: after sunrise (from 6:00 to 9:00), during the daytime (from 9:01 to 13:00) and before sunset (from 17:00 to 21:00). Hours refer to CEST.

| Site | 2017 | | | 2018 | | | Total |
|------|------|------|------|------|------|------|-------|
| | After Sunrise | During Daytime | Before Sunset | After Sunrise | During Daytime | Before Sunset | |
| MOL | 0 | 6 | 8 | 7 | 4 | 17 | 42 |
| SP1 | 3 | 7 | 5 | 10 | 6 | 15 | 46 |
| SP2 | 3 | 3 | 5 | 9 | 1 | 11 | 32 |
| Total | 6 | 16 | 18 | 26 | 11 | 43 | 120 |

During the two consecutive summers, from late June to August, reproduction-related behaviours were observed on 12 occasions (*n* = 5 in 2017, *n* = 7 in 2018) only at MOL. Gatherings of *M. rubra* ranging from 6 to 30 individuals were observed nine times (15 ± 9 individuals, mean ± ES; *n* = 3 in 2017, *n* = 6 in 2018) during evening hours in seven out of the nine observations and no other fish aggregation was seen co-occurring. The largest aggregations were observed twice, on 12 July 2017 and 21 August 2018, between 18:30 and 20:10. Each aggregation counted 20–30 individuals of *M. rubra*; individuals were aggregating in the water column (roughly at 18–25 m depth) close to the seaward end of the highest rocky ridge (20–30 m depth range) characterising the MOL site (Supplementary Materials Video S1). The aggregations were observed from a distance of ~3–10 m, as individuals tended to disperse when approached by SCUBA divers. Although the greatest care was taken not to alter fish behaviours, aggregated grouper might have been disturbed by the presence of divers [17,78]. Within these gatherings, individuals were displaying the three colour patterns known from the literature (mottled, silver and dark patterns; Figure 5A,B). Presumptive courtship displays were performed by fish with lengths ≥60 cm (TL) and displaying the silver pattern. Specifically, silver individuals were observed rising in mid-water in an almost vertical position by means of the rapid contraction of their lateral musculature and then descending along an arched path towards the bottom (Supplementary Materials Video S2a). Courtship behaviours were observed both within and near aggregations. Silver individuals generally targeted grouper with the uniform dark pattern (Figure 5B, Supplementary Materials Video S2b).

While conducting behavioural observations at MOL, considering both summers, the mean seawater temperature recorded by the dive computer was 21.8 ± 2.7 °C. Similar values were retrieved from the temperature sensor, which recorded an average seawater temperature of 22.5 ± 2.9 °C from 1 August to 10 September in 2018.

### 3.3. Acoustic Recordings

Two sound types, mostly recorded in pairs, were only found in the recordings made at the fully protected site of MOL. The recorded signals have never been reported to date [75,79,80] and were likely emitted by a grouper species as they share typical acoustic features, such as low-frequency pulse series often combined with a downsweeping sound [34–39,46]. In the TPCCMPA, other grouper species and documented soniferous fish species have been censused (Supplementary Materials Table S2; [67,75,81]). However, the dusky grouper was the only other grouper species recorded at MOL concomitantly with the present study [65] and its calls as well as the ones emitted by other known vocal fish occurring in TPCCMPA are distinct from the sounds described here [27,46,75,82–85]. The

presumed mottled grouper calls were referred to as a low-frequency fast pulse train (LFPT; Table 3, Figure 6) often combined with a downsweeping sound (DS; Table 3, Figure 6). On average, LFPT sounds had a duration of 477 ± 122.5 ms (mean ± SD) and were composed of 9.1 ± 2.7 pulses (19 ± 2.8 pulses s$^{-1}$), with an IPI of 59.1 ± 12.5 ms and a peak frequency of 82.2 ± 35.7 Hz (Table 3). DS sounds lasted 337.1 ± 164.2 ms on average and had a mean peak frequency of 216.6 ± 70.3 Hz (Table 3).

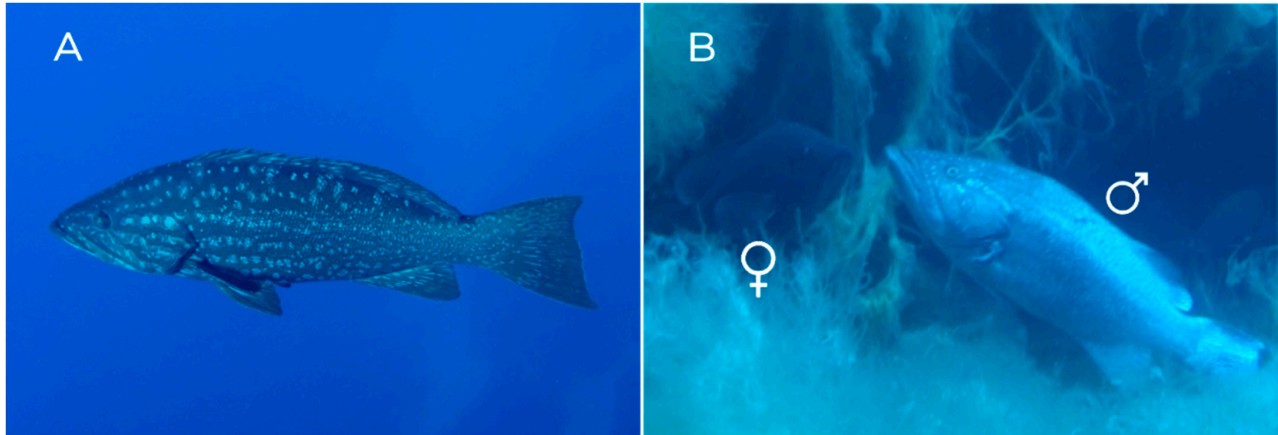

**Figure 5.** Pictures of three individuals of *M. rubra* displaying the three colour patterns known from the literature: (**A**) basic mottled pattern, (**B**) dark uniform pattern (presumptive female) and silver pattern (presumptive male). The behaviour displayed by the presumptive male in B is available in the Supplementary Materials Video S2b. Note, in B, the proximity of *M. rubra* individuals to the bottom, covered by mucilaginous aggregates, likely of the algae *Acinetospora crinita*.

**Table 3.** Descriptive statistics of the two sound types likely produced by *Mycteroperca rubra* during courtship. Temporal measurements (duration and inter-pulse interval, IPI) were performed on the oscillogram of the sounds, while frequency parameters were measured on the spectrogram (LFFT size = 256, Hann window, window overlap 50%); frequency 5% and frequency 95% are the frequencies dividing the sound selection into two frequency intervals containing, respectively, 5% and 95% of the energy in the selection; bandwidth 90% is the difference between the 5% and 95% frequencies.

| Sound Type | Variable | Minimum | Mean | ±SD | Maximum | *n* |
|---|---|---|---|---|---|---|
| Low-frequency Fast Pulse Train (LFPT) | Duration (ms) | 242.0 | 477.0 | 122.5 | 817.0 | 43 |
| | Pulse sound$^{-1}$ | 4.0 | 9.1 | 2.7 | 15.0 | 43 |
| | Inter-pulse interval (IPI, ms) | 42.0 | 59.1 | 12.5 | 131.0 | 347 |
| | Peak frequency (Hz) | 30.5 | 82.2 | 35.3 | 219.1 | 87 |
| | Frequency 5% (Hz) | 0.0 | 43.7 | 16.0 | 62.6 | 87 |
| | Frequency 95% (Hz) | 62.6 | 180.4 | 45.4 | 313.0 | 87 |
| | Bandwidth 90% (Hz) | 0.0 | 136.8 | 53.2 | 281.7 | 87 |
| Downsweeping Sound (DS) | Duration (ms) | 120.0 | 337.1 | 164.2 | 1241.0 | 58 |
| | Peak frequency (Hz) | 45.8 | 216.6 | 70.3 | 427.2 | 86 |
| | Frequency 5% (Hz) | 45.8 | 140.0 | 34.8 | 198.4 | 86 |
| | Frequency 95% (Hz) | 244.1 | 319.4 | 59.6 | 442.5 | 86 |
| | Bandwidth 90% (Hz) | 61.0 | 179.4 | 72.3 | 381.4 | 86 |

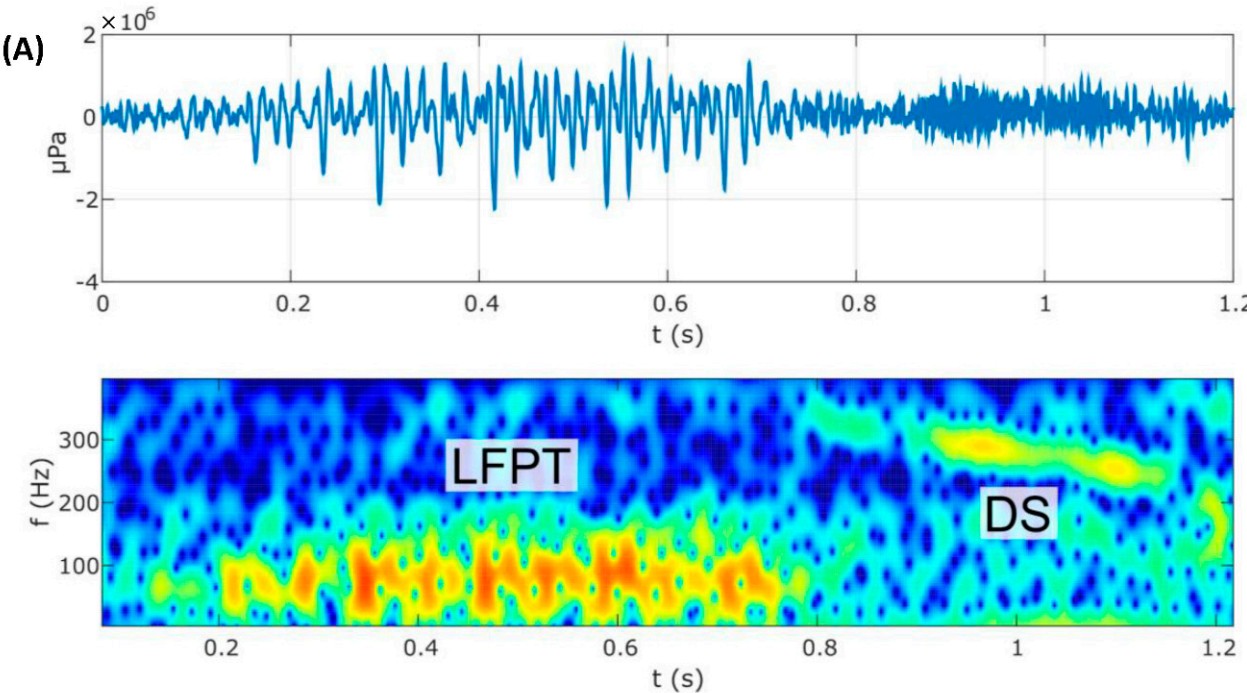

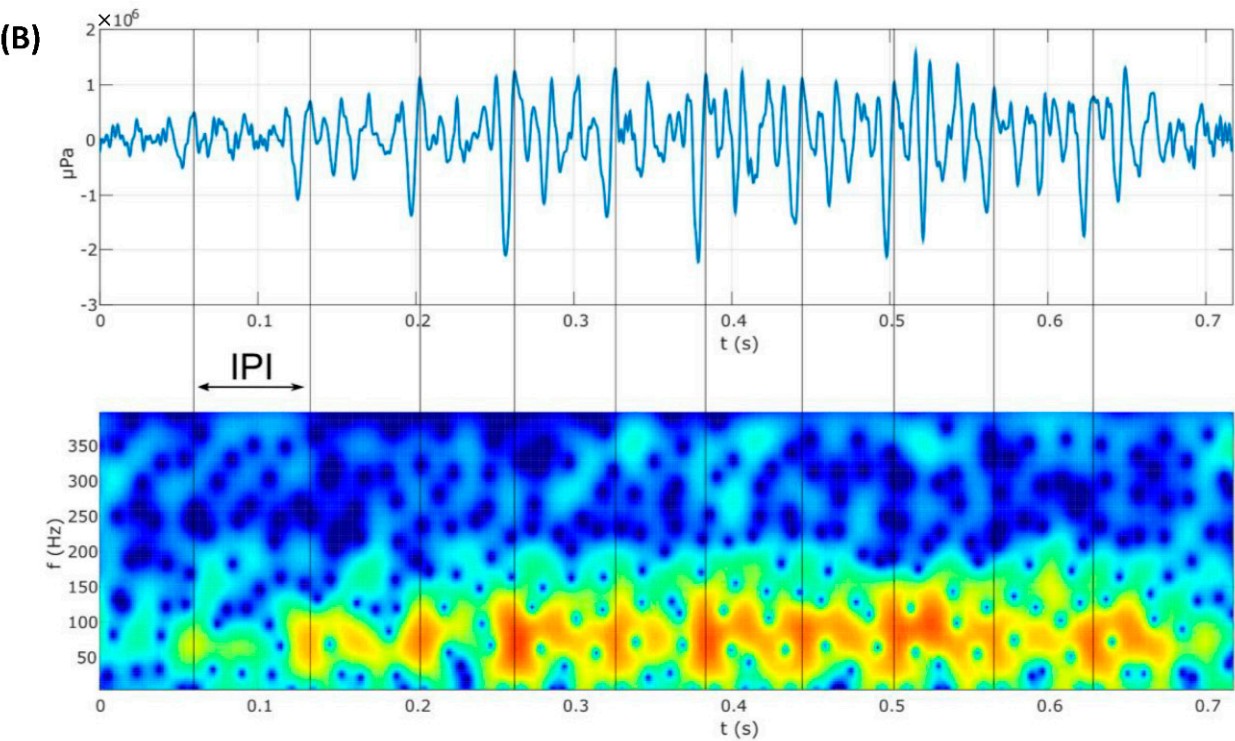

**Figure 6.** (**A**) Oscillogram (**top**) and spectrogram (**bottom**; fast-Fourier-transform (FFT) length = 256 samples/points, Kaiser window) of the two sound types presumably produced by the mottled grouper. LFPT: low-frequency fast pulse train; DS: downsweeping sound. The corresponding audio file is available in the Supplementary Materials. (**B**) Zoomed oscillogram (**top**) and spectrogram (**bottom**; fast-Fourier-transform (FFT) length = 256 samples/points, Kaiser window) of the LFPT sound. Vertical lines indicate pulse peaks (10 pulses). The inter-pulse interval (IPI) is the peak-to-peak interval and the time span from the first pulse peak to the last pulse peak was used to estimate sound duration.

### 3.3.1. Temporal Patterns in Sound Production

At MOL, around 44 LFPT and 49 DS sounds min$^{-1}$ were recorded throughout July 2017. In contrast, no calls were recorded at the rocky banks (SP1 and SP2) during the same period. However, the acoustic sampling effort at the three sites was not the same, with around 180 effective hours of recording collected at MOL, 89 at SP1 and 78 at SP2 (Supplementary Materials Figure S2). Audio recordings overlapped at the temporal scale between study sites and were more continuous at MOL throughout the month, as shown in Figure 7. At MOL, the grouper-like sounds were more often detected between 18:00 and 21:00 (Figure 7). A peak in sound production occurred for both LFPTs and DSs on 12 July 2017, exactly when the putative spawning aggregation and courtship behaviours of *M. rubra* were documented (Supplementary Materials Videos S1 and S2). There were fluctuations in sound production, but the temporal and abundance trends were the same for both sound types throughout the month. Considering the acoustic data collected from 18:00 to 8:00 only, mean LFPT sounds were $0.12 \pm 0.05$ min$^{-1}$ (mean $\pm$ SD), while mean DS sounds were $0.13 \pm 0.09$ min$^{-1}$.

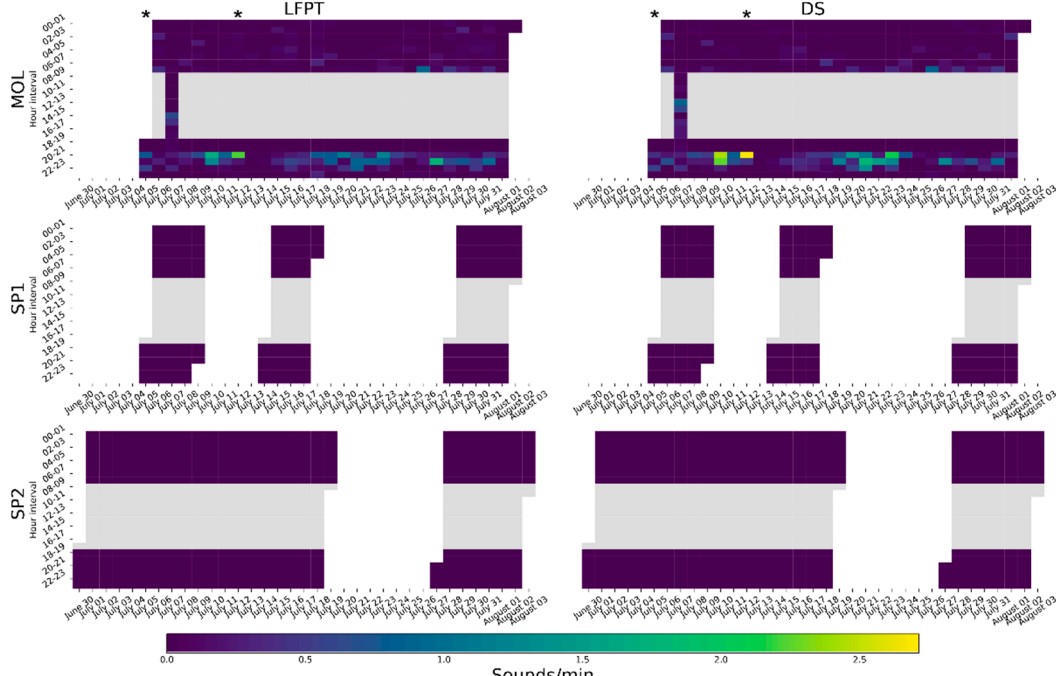

**Figure 7.** Weighted sound abundances (number of sounds by sound type (left: LFPT; right: DS) per effective minute of recording within an hour interval) recorded across the three sites throughout the month of July 2017 (hours refer to CEST). The asterisks (*) indicate the dates when presumptive courtship displays and also an aggregation (12 July) were observed. The white areas indicate absence of acoustic data, while grey areas indicate that recordings were not analysed. On 7 July, the recordings for the whole day (24 h) were analysed.

### 3.3.2. Relationship between Visual Observations and Presumed Acoustic Behaviour

In summer 2018, indirect signs of reproduction, such as changes in colour pattern and courtship behaviours, were observed during seven dives performed in July–August. However, concurrent audio recordings were only available for two dives during which spawning aggregations and courtship behaviours were observed. Consequently, no statistical tests could be conducted. Overall, 11 NC ("No-Courtship") dives ($n = 2$ after sunrise, $n = 5$ during daytime, $n = 4$ before sunset) and 2 C ("Courtship") dives (before sunset) were analysed. On average and per effective hour of recording, $50.3 \pm 21.1$ LFPT (mean $\pm$ SD) and $46.2 \pm 11$ DS sounds were detected during C dives, and $7.6 \pm 7$ LFPT and $10.2 \pm 11.1$ DS sounds during NC dives, suggesting a higher level of vocal activity during courtship.

## 4. Discussion

This study reveals novel insights into the reproductive biology of *M. rubra* and identifies a new presumptive spawning location in the western Mediterranean Sea. Animal density, size, colour pattern and behavioural data suggest that *M. rubra* reproduces within the TPCCMPA, especially in the surroundings of Molarotto island (MOL), and likely emits two call types found to be more abundant when reproductive behaviours were observed in the field. These two call types have never hitherto been described for this species.

A clear difference in the density and size-frequency distributions of *M. rubra* was found across the three study sites. Density was greater at MOL than at SP1 and SP2 and this may be due to (i) the topography of the sites, (ii) the higher number of grouper belonging to other sympatric species that may compete for resources [65] and (iii) the anthropogenic disturbance, especially due to recreational diver frequentation, which is relatively intense at SP1 and SP2 but absent at MOL. *M. rubra* occurs over rocky and adjacent sandy bottoms and it is generally more common down to 30 m depth [51,53,86]. Although all study sites are characterised by rocky reefs rising from sandy bottoms, MOL is overall shallower, and slopes are gentler than at SP1 and SP2, thus corroborating previous findings indicating a preference of *M. rubra* for gently sloping rocky plateaus [51]. *M. rubra* depth distribution overlaps with that of *E. marginatus* [51], suggesting they may compete for shared resources (e.g., shelter, food, spawning territories), especially at SP1 and SP2, which were reported as potential spawning aggregation sites of *E. marginatus* [65,67]. However, competition between these sympatric species seems unlikely considering that *E. marginatus* shows a bottom-dwelling behaviour with a preference for shelter-rich rocky reefs (peculiar to SP1 and SP2), while *M. rubra* exhibits more pelagic habits with a preference for less uneven, rocky substrates (distinctive of MOL) [51,86]. Moreover, no evidence is available concerning competition for food resources [87].

Anthropogenic disturbance due to diving activity could also have an impact, as the two species may react differently to the divers' presence. While MOL (A Zone) is only occasionally visited by scientific divers, SP1 and SP2 (B Zone) are popular dive sites attracting more than 10,000 recreational divers each year [88]. Some authors described *M. rubra* as more sensitive [52,55] and others as less suspicious with respect to SCUBA divers than other grouper species [51]. Additionally, the mottled grouper might be increasingly more wary towards divers as protection level decreases and fishing pressure increases. In fact, despite being closed to fishing, the study sites fall in different protection zones and the detectability (detection distance) of this species during visual census surveys has been found to decrease with decreasing levels of protection, moving from fully protected areas to unprotected areas [89].

*M. rubra* displayed larger sizes at MOL than at SP1 and SP2, suggesting that MOL is more suitable for the large-sized individuals (i.e., spawners) of this species. *M. rubra* is a protogynous hermaphrodite, changing sex from female to male [57]. At the three study sites, all censused individuals were potentially sexually mature adults, with sizes ranging from 45 to 85 cm (total length, TL). Although the sexes of individuals could not be determined, the smallest recorded length was greater than that at which females are reported to attain sexual maturity (35 cm TL [52,57]). Moreover, based on the reported sizes of female-to-male sex change (40–65 cm TL [52,57]), it is reasonable to suggest that individuals longer than 65 cm TL, which were only recorded at MOL, were sexually mature males. The in situ observations further strengthen this hypothesis, as individuals displaying the silver pattern and performing courtship behaviours, likely males [55], were at least 60 cm long (TL).

Field observations allowed documentation of large gatherings of *M. rubra* (<30 individuals) during evening hours only at MOL. During such unusual gatherings, indirect signs of reproduction (colour changes associated with courtship behaviours) were collected, suggesting they were spawning aggregations [52,53,55,56]. Characterising such aggregations in terms of density and size-frequency distributions would require conducting UVC surveys in the evening, since the density of aggregating fish is known to change during the same day [78]. Moreover, evening was also the period when higher sound

abundances were recorded. In fact, this study provides novel evidence that *M. rubra* likely emits courtship-associated sounds. Using passive acoustic monitoring (PAM), two distinct sound types were only recorded at MOL. These calls, referred to as low-frequency fast pulse trains (LFPTs) and downsweeping sounds (DSs), share the typical acoustic features of grouper courtship calls (frequency range and composition in pulses, and down-sweeping nature) but are distinct from the calls of *E. marginatus*, the other grouper species observed at MOL [46,75]. Peak sound production occurred on 12 July 2017, when a gathering of *M. rubra* was concomitantly observed, thus suggesting an increase in the intensity of courtship activity and in fish density. Moreover, LFPT and DS sounds were detected throughout July 2017 during crepuscular and early night-time hours, following a similar trend in abundance. This evening sound production was already documented for other groupers displaying reproductive behaviours [34–39,46]. The sequence of movements exhibited during the courtship display of the mottled grouper could be related to the emission of the LFPT and DS sounds: the rapid contraction of the lateral musculature during the rise might be associated with the production of the quickly repeated low-frequency pulses (LFPTs) as observed for other grouper species [39], while the following descent towards the bottom might be associated with the production of the downsweeping sounds (DSs). When comparing the number of caudal fin flaps measured during one *M. rubra* shaking-like event (Supplementary Materials Video S2a) and the average number of pulses in LFPT sounds, relatively close estimates were found: 14 caudal fin flaps s$^{-1}$ versus 19 ± 2.8 (mean ± SD) pulses s$^{-1}$. It is therefore reasonable to believe that the contraction of the lateral musculature associated with caudal flapping may lead to the emission of sounds. Further studies are needed to confirm the potential link between the number of caudal fin flaps and the number of emitted pulses.

Although a direct link between sound production and the emitting species could not be assessed, all these elements concur to strongly suggest that these newly described sounds are produced by *M. rubra*. To confirm this association, further studies should include the use of an underwater camera coupled with a synchronised hydrophone [34,36,37,39,77]. Fixed underwater cameras, being less intrusive than cameras carried by divers, would be more appropriate in identifying *M. rubra* species-specific sounds.

The largest aggregation of *M. rubra* ever recorded consisted of about 500 individuals/200 m$^2$ and occurred off the coast of Israel, within the marine reserve of Rosh Hanikra [52]. In our study, individuals were observed aggregating only at MOL, close to the most seaward end of the highest ridge characterising this site, which shares topographical similarities with the Israeli location [52]. The MOL site also falls within the most remote fully protected zone of TPCCMPA. Therefore, the observed reproductive activity of *M. rubra* may benefit not only from the topography and the remoteness of this site but also from the protection from fishing. At the Mediterranean scale, effective protection has been found to provide the most significant benefits to fish assemblages associated with offshore structurally complex habitats with high hydrodynamism [7,90]. It has also been shown that fully protected areas increase grouper biomass via the protection of large-sized individuals (i.e., spawners) [5,67].

Multiple Mediterranean grouper species, such as *E. marginatus* and *M. rubra*, are known to aggregate for reproduction at particular sites (i.e., multi-species spawning sites), such as rocky banks [51,67]. There is little indication that other groupers share the same spawning sites as *M. rubra* at different times. Aronov and Goren [52] reported that towards the end of the spawning season of *M. rubra*, *E. marginatus* and *E. costae* were observed aggregating, likely for spawning, exactly where the densest aggregations of *M. rubra* occurred. Observations on the courtship activity of *E. marginatus*, conducted in TPCCMPA concomitantly with this study, indicated that the MOL site and its surroundings might be a spawning ground for both *M. rubra* and *E. marginatus* during summer, with spawning potentially occurring on different days and/or at different times [66]. As shown elsewhere, spatiotemporal partitioning in the use of the same spawning sites by multiple groupers could be explained by the competition for limited spawning habitats [17]. Literature data

showed that PAM can be extremely useful in discriminating between different grouper species gathering at multi-species spawning sites at different times [40,41]. Specifically, using PAM at these sites would result in the monitoring of reproduction-related dynamics of multiple species over time [17,28].

This study provided ample evidence of *M. rubra* aggregation and courtship behaviours within TPCCMPA. Specifically, the observation of aggregating fish, courtship behaviours and associated colour patterns at the site of MOL suggests that it may serve as a courtship arena or an actual spawning site of *M. rubra* [91]. Assuming that MOL is a courtship arena implies that spawning may have occurred in its close surroundings because, by definition, courtship arenas of transient aggregating species are known to immediately surround spawning sites [91]. Additionally, rocky reefs, the suitable habitat for this species, become increasingly patchy with increasing distance from the islet of Molarotto, providing a further indication that spawning may have occurred near MOL.

The present research will serve as a basis for future investigations aimed at validating *M. rubra* sound production and the benefits of protection on the occurrence of aggregations. Similar studies should be conducted in other MPAs reporting the presence of *M. rubra*. Specifically, if the courtship-associated sounds of *M. rubra* are validated, this study will support the use of PAM for the identification of its aggregation spawning sites within MPAs and the monitoring of the spatiotemporal patterns of its reproductive behaviours. Additionally, the use of PAM would help locate multi-species aggregation sites where also the soniferous and endangered *E. marginatus* aggregate to spawn. Such data are crucial to inform managers about the sites/areas that warrant further enforcement efforts and ultimately support the implementation/expansion of fully protected areas within multiple-use MPAs. This would help improve the conservation and fishery outcomes of individual MPAs or networks of MPAs [4,8,9] and, at a wider scale, ultimately contribute to the more consistent achievement of international targets (e.g., the "30 × 30" goal [4,14]).

**Supplementary Materials:** The following supporting information can be downloaded at: https://www.mdpi.com/article/10.3390/d14050318/s1, Table S1: *Post hoc* analysis by Dunn's test, Table S2: Alphabetical list of the 19 fish taxa visually censused in the study area, Figure S1: Sampling schedule of diving activities, Figure S2: Sampling schedule and recording cycles, Figure S3: Custom-built MATLAB interface, Video S1: Aggregation of *M. rubra*, Video S2: (a) Presumptive courtship display by a silver grouper. (b) Presumptive courtship display by a silver grouper toward a dark grouper (presumptive female), Audio file: Suspected courtship associated call of *M. rubra*.

**Author Contributions:** Conceptualization, E.D., C.M., P.G. and L.D.I.; methodology, E.D., C.M., P.G. and L.D.I.; software, C.G. and L.D.I.; formal analysis, E.D. and L.D.I.; investigation, E.D., P.P., P.G. and L.D.I.; resources, C.G. and L.D.I.; data curation, C.G. and L.D.I.; writing—original draft preparation, E.D.; writing—review and editing, E.D., C.M., A.N., P.P., C.G., P.G. and L.D.I.; visualization, E.D., C.G. and L.D.I.; supervision, C.M., P.G. and L.D.I.; project administration, E.D., C.M., A.N., P.G. and L.D.I.; funding acquisition, A.N., C.M. and P.G. All authors have read and agreed to the published version of the manuscript.

**Funding:** This research was carried out within the framework of the PhD project of E. Desiderà, which was funded by the Marine Protected Area of Tavolara–Punta Coda Cavallo, Ministero della Transizione Ecologica (MiTE, previously named Ministero dell'Ambiente e della Tutela del Territorio e del Mare, MATTM). This research was also supported by funding from the: (i) Ministero dell'Istruzione, dell'Università e della Ricerca (MIUR), (ii) Consorzio Nazionale Interuniversitario per le Scienze del Mare (CoNISMa) and (iii) Istituto Superiore per la Protezione e la Ricerca Ambientale (ISPRA) in the framework of the project dealing with the "health status of coastal fish communities", in application of the Marine Strategy Framework Directive (MSFD).

**Institutional Review Board Statement:** Not applicable.

**Informed Consent Statement:** Not applicable.

**Data Availability Statement:** The data analysed in this study are available on request from the corresponding author with the permission of the Marine Protected Area of Tavolara–Punta Coda Cavallo.

**Acknowledgments:** We thank Luana Magnani and Egidio Trainito for their invaluable assistance in the field. We also thank Rémi Blandin for helping with MATLAB coding. We wish to thank Michael Paul for the English language revision.

**Conflicts of Interest:** The authors declare no conflict of interest. The funders had no role in the design of the study; in the collection, analyses, or interpretation of data; in the writing of the manuscript, or in the decision to publish the results.

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
