# Peer review of "Reproductive Behaviours and Potentially Associated Sounds of the Mottled Grouper Mycteroperca rubra: Implications for Conservation"

_diversity, doi:10.3390/d14050318_

Round 1
Reviewer 1 Report
This study is well researched and presented but I have the following general concerns that will make it difficult for this work to be published in 'Diversity' in its current form.
1) The methods used here are very much standard and there is very little innovation. For example, in the discussion on page 429-430, it is suggested to use cameras coupled to hydrophones to identify the caller. Yet, this technology is available since man years and this begs the question, why the authors did not apply it in the first place.
2) The study is very descriptive, and it does not seem that the authors have any hypothesis to test. The aim seems to have been to just document the patterns of a fish species that might be targeted by fisheries in the future. To underline this, the authors say in the abstract that 'this study provides a basis for advancing our knowledge about M. rubra reproductive biology'. This is a very narrow objective and thus the overall biological significance seems very low. This is underpinned also by the relatively small study area.
Given this general impression, I recommend the authors substantially revise their work to formulate a relevant hypothesis and justification for the study. I give some advice further below. Alternatively, the authors might consider submitting their works to a lower ranking scientific journal, perhaps covering the Mediterranean area specifically. Don't get me wrong: it looks like a fine and dedicated job but the potentially interested readership seems small.
Some specific remarks:
Introduction, line 39-41. The authors start with a very relevant study objective when saying: To know where and when marine fishes of ecological and commercial importance reproduce can provide key information for their stock management and conservation, ultimately guaranteeing the viability of their populations. Yet, they do not specify how the biological information is used in marine conservation exactly. Thus, their statement becomes very generic, basically saying that biological knowledge can be used in conservation. They then go on describing what we know about the life history and distribution of groupers, and they also identify the open question they will investigate (line 92-97). However, 'novel insights' and researching something 'for the first time' are simply not sufficient enough to justify a scientific study. The authors should identify what information can be used in marine protection, how this is used precisely and then how such information can benefit their study species which seems currently not under threat. I am sure, such information is out there. But just implying that a study is useful because 'nothing is none about' is not enough.
Discussion, line 359-364: Following their general assumption of finding something new is always good, the authors inform us that they (maybe) have identified a new spawning area for their species in the Med. Yet again they fail to explain, why this is important information. Isn't their study site already a marine protected area, which indicates that it is important for fishes? And how many other spawning areas have been described. In the same manner, the authors state that they found two call types that have never been described for their study species. So what? Why is this information important? How can it be used?
Discussion, line 459-462: Finally, and in the very last paragraph of the paper, we find a statement which attempts to explain the whole point of the study: Similar data are fundamental for guiding conservation/management actions, as protecting the reproductive sites of multiple exploited species, such as groupers, can significantly increase their reproductive potential, and help rebuild depleted populations [44,60,61]. Such data also help improve the bio-ecological benefits of individual MPAs and, at a wider scale, more consistently achieve international targets.
The first part is quite precise but again, isn't the site already under protection? The second part is OK, although one could have been a bit more specific, e.g., what is meant by 'improving biological benefits of MPAs'? and what targets (MSFD) are referred too? Yet, my point is that the authors should start the paper with this section because it provides context and motivates the reader to carry on reading. Thus, I guess, my suggestion is to start the paper from the end using the above statement as a starting point.
Reviewer 2 Report
General impression
The paper provides a first description of combined visual and acoustic activity of the mottled grouper and suggests that passive acoustics is an interesting tool for long term monitoring of groupers’ spawning areas. This is therefore a nice contribution to the field of bioacoustics and merits publication in the Marine Diversity Journal. However, I believe that several points need to be addressed to clarify some methodology items, results/discussion statements and make the manuscript easier to read.
Below is my detailed list of comments and suggestions on the Methods, Results et Discussion sections
- Methods
181-182 : list all the metrics measured on the sound extracts (Duration (ms), Inter-pulse interval (IPI, ms), Peak Frequency (Hz), Frequency 5% (Hz), Frequency 95% (Hz), Bandwidth 90% (Hz), Number of pulses)
171 : Recording cycles differed among sites depending on the recording devices used and the duty cycles : can you describe recording periods for each site and the duty cycle ?
173-181 : examples of spectrograms of the known dusky grouper and other grouper calls would be nice to visualize the type of sounds the analysts were looking for.
190-192 : Can you indicate the list of sound types ?
- Results :
232-234 : 2 identical sentences.
228 : Indicate that “ individuals 125 m2” is the transect area
Figure 2 : I do not quite understand how the transects were carried out while diving. Can you add in Figure 2 the number of transects carried out for each date within each site (Number of replicates (belt transects) per site per date)
242-244 : Specify whether the test was carried out in 2018 only or both years. If the 2017 dataset was considered, it seems a bit limited to compare different sites over different years (especially given the inter-annual differences, cf, Fig 2)
Figure 3 : it seems a bit limiting to compare different sites over different years. Especially given the inter-annual differences illustrated in Figure 2
271 : “ranging from 6 to 30 individuals” : this is quite interesting so it would be nice to have more information about this result – at least mean (+/-std) or median/Q1/Q3.
279 : “individuals tended to disperse when approached by SCUBA divers” – can you add a sentence presenting the intrusive aspect of diving as a limit of the study (bias for the estimation of densities). It is mentioned in the discussion, but I think it can also be pertinent in the results section.
340 : ​​“Considering daily 14-hour intervals, mean LFPT sounds were 0.12 ± 0.05 min-1 (mean ± SD) while mean DS sounds were 0.13 ± 0.09 min-1 “ : this sentence should be better explained.
353 : “NC dives / C dives” – can you explain ?
Additional comment:
It would be relevant to indicate the temperature of the water since this parameter is likely to influence muscle contraction and therefore sound production (already seen in other species - decrease in sound frequency with temperature).
- Discussion
378 : “especially at SP1 and SP2, which represent potential spawning aggregation sites of E. marginatus [44]” => I don’t understand this conclusion given the information about the sounds of E. marginatus grouper reported in this study (did sounds of E. marginatus grouper were detected in this study ?).
429 : To confirm this association, further studies should include the use of an underwater camera coupled with a synchronised hydrophone -> Tank studies could also be useful, even if it involves biases in behaviour and sound measurement. It would be easier to obtain simultaneous acoustic / video data in a tank study (especially since the reproduction episodes seem very punctual, therefore difficult to manage using in situ video acquisition level).
451-460 : “[Observations on the reproduction of E. marginatus, … Specifically, using PAM at these sites would result in the monitoring of spawning dynamics of multiple species over time] “. => I don’t believe this behaviour has been observed in this study by divers or acoustic data?
432-444 : I suggest moving the whole paragraph after ligne388

Reviewer 3 Report
The manuscript by Desidera et al. combines an extensive set of underwater visual census surveys with PAM analysis to assess the presence and reproductive behaviour of a common but relatively understudied grouper species within a Mediterranean MPA. The combination of methods provides some useful insights on the species' reproductive biology along with implications and future possibilities for MPA management. Overall, the methodology and presentation of results is sound, although one could consider a recurrent weakness the lack of a direct proof of association between the vocal signal recorded and the specific grouper species (M. rubra). This results in a widespread presumptive sense along the manuscript. I think this could be overcome by the authors providing some supporting evidence for their hypothesis, e.g. by listing the presence (or absence) of other vocal fish species in the area and highlighting the differences of the presumed M. rubra signal over that of the other species.
More specific comments and suggestions are provided below:
L64 to 69: I would add that apart from fishing, habitat degradation, shift and loss due to anthropogenic and exogenous pressures is a major future threat for Mediterranean biodiversity, including groupers, as supported by many recent studies.
L115 to 120: I missing here some information on the topographic features of the three surveyed sites. What is their bathymetry? inclination/rugosity? prevailing habitat? Are these features similar or differ between the three sites?
L122: were the three divers conducting UVC simultaneously?
L122: Better replace “UBCs” with “UBC surveys”
L124 to 125: Just a clarification: I guess four distinct replicate strip transects randomly placed over the surveyed area were conducted, right? (As opposed to four replicate observations over a single strip transect).
L168 to 169: continuous recording over extended time periods is also reported in Fig. S2 at SP1 in 2017 and SP2 in 2018. By the way, why are there two series of recordings at SP1 in 2018 (one continuous and several short-span rotations)?
L170 to 171: Why is that and why is it mentioned here? Is there some temporal non-overlap between locations? Better just refer to the number of recording samples acquired from each location here or in the Results.
L227 to 234: Correct duplicate paragraph and possibly refer to mean (SD) density values for all locations/years.
L218 to 234: were there other groupers (or known vocal species) observed at the studied locations during the transects?
L316 to 327: I suggest merging these two figures into one (a and b) since they actually refer to the same sound type. Or, alternatively, move Figure 7 to the supporting information, since it is not conveying much new information elaborated in the Results text. If the point is to better show the LFPT features, then you could annotate the image, i.e. with symbols over each pulse peak and IPI and a double arrow indicating the total time span of sound duration.
L319: replace “that might be produced” with “presumably produced”
L335 to 337: Could the authors add the time points of courtship behaviour observations at MOL in Figure 8, so that it could be demonstrated how they are distributed in relation to sound observations?
L361: replace “could reproduce” with “reproduce”
L362: however, no evidence for reproduction of the species was revealed in the other two locations
L368 to 369: If there are other groupers reported for the studied locations (which would be expected), I think the authors should make a reference to them (species, densities), either citing their own data or other published information.
L375: replace “plateaux” with “plateaus”
L404 to 405: However, this is not elaborated in the results, i.e. evening sound recordings were selectively analysed on the presumption that this is the time of day with the most vocal activity and less surrounding noise. Did the authors perform a preliminary check on the data? Moreover, I recommend the authors comment on the fact that visual observations were performed during daytime and acoustic observations during nighttime. Is this discrepancy expected to affect the results and derived conclusions?
L406 to 411: some repetition here with the corresponding passage in the Results
L439: at <the> Mediterranean scale
L450 to 453: I am missing here some comment on the possible recording of E. marginatus vocal signals in the studied areas? Were there any? If so, how confident are the authors for the discrimination of these signals over those of M. rubra?
Reviewer 4 Report
This manuscript aims to describes the behavior and vocalizations associated with spawning of the mottled grouper, M. rubra. Additionally, the authors examine the density and size distribution of individuals of this species among three sites. The study and data presented provide novel information on sound production in M. rubra. The acoustic data presented is an important contribution to the literature as it is the first description of sound production in this species. Likewise, the information on the spawning associated color patterns and behaviors are also interesting. However, there are multiple areas that require major revisions before the manuscript can be properly evaluated. More details are needed in the methods to accurately understand how the data was handled. There appears to be a problem with sub-samples used as independent replicates with the underwater visual transect data, which confounds multiple aspects of the analysis. Additionally, more background information is needed regarding dynamics of spawning aggregations and spawning behavior to put this research in context of existing literature.
General Comments:
Introduction:
Since there appears to be very little previous information on the target species M. rubra, the reader would benefit from seeing more examples of spawning dynamics, behaviors, and associated vocalizations of related species. That will help set up the premise of this study and also provide a basis to compare the results of this study.
Methods:
The inclusion of the SP1 and SP2 study sites is questionable. I do not see the value they provide as such little data is derived from them. Also, It is not clear that they actually represent two distinct sites. Additional information on the UVC transects is needed to evaluate the quality of the data (i.e. how far apart were transects, were they parallel or perpendicular, how did you avoid counting the same fish more than once? Etc. ). As mentioned above, I do not believe that the individual transects represent independent samples, and as such all related statistical analysis should be revised.
Specific Comments:
Line 21: “nowadays” odd word choice, please clarify distinction between what is the only documented sound producer and setting up your study.
Line 29: “hitherto” also odd word choice. Make more concise
Line 39: change “To know” to “knowing”
Line 48-52: Here, I would like to see more specific information about the range of spawning behaviors, especially with regards to aggregating behavior. (See Biggs CR, Heyman WD, Farmer NA, Kobara S, Bolser DG, Robinson J, Lowerre-Barbieri SK, Erisman BE (2021) The importance of spawning behavior in understanding the vulnerability of exploited marine fishes in the U.S. Gulf of Mexico. PeerJ 9:e11814.)
I would also like to see more examples of courtship or spawning behaviors that have been linked to vocalizations. For example: Rowell TJ, Aburto-Oropeza O, Cota-Nieto JJ, Steele MA, Erisman BE (2019) Reproductive behaviour and concurrent sound production of Gulf grouper Mycteroperca jordani (Epinephelidae) at a spawning aggregation site. J Fish Biol 94:277–296.
Line 54: “scant”
Line 55-56: Since E. marginatus is the most studied, is their information on the vocalizations of that species or on the behavior associated with vocalizations? That information would help set up comparison for your results.
Line 60-61: incomplete sentence “because they are thermophilic”
Line 71-72: You already said this on line 62
Line 82: I think a description of “particular behaviors” is warranted and will help the reader understand the likely courtship behaviors.
Line 88-90: It would be helpful to introduce more information about the life history of M. rubra here. What is the size at maturity, the mating system, and sexual pattern? At least, a further description of the Aronov (2008) study is needed.
Line 117: From the figure it looks like SP1 and SP2 are the same site. How far apart are those sites, and how likely is it that two spawning sites would be that close together for an aggregative spawner? I believe incorporating further explanations of aggregative spawning dynamics (as described in Biggs et al. 2021) will help clarify those distinctions.
Line 124-125: It sounds like each of the 4 transects per survey are not independent of each other, but it is not clear how that data was handled. The transects should be considered sub-samples of each dive or day per site. Additionally, with random transects, how are you avoiding counting the same individual fish more than once?
Line 130:
Line 132: How was density at each site calculated? Are each transect considered independent?
Line 144: rather than livery I think it would be better to mention the specific color patterns (mottled, dark, silver)
Line 144: What constitutes courtship activity? This should be described in the intro more specifically and probably mention it here as a reminder.
Line 151-153: Yes, the contraction of the lateral musculature may provide evidence that the fish is the source of the sound being recorded, but I am not aware of any research supporting the idea that the number of tail flaps or lateral muscle contraction speed has any relationship to the characteristics of the sound being produced.
Line 169: what were the duty cylces or at least the range of duty cycles? I know it’s in the supplemental, but I think it would help the reader to see that information here.
Line 174-175: do you mean a low-pass filter was applied to remove higher frequencies? Some clarification is needed to understand why they were down-sampled.
Line 190: “us to select”
Line 194: I think it would help to clearly lay out the criteria for deciding what constitutes a spawning aggregation. As I mentioned before, there are a spectrum of aggregating behaviors, so making this distinction clear is important.
Line 210-204: I think this is okay, but sounds a little weird and convoluted to me, is it not just the average vocalization per minute of recording?
Line 221: Based on the details provided in the methods, I am doubtful that each transect can be considered an independent sample, and that fish were not counted multiple times. For example, were the 3 transects with M. rubra at SP2 on the same day?
Line 246/Figure 3: Again, based on the information provided, I don’t believe each belt transect to be a true replicate as they are not independent samples. This should be fixed and the stats run again.
Line 270: Since spawning behavior is only recorded at MOL, what is the benefit of including the other two sites in the analysis other than as perhaps a reference of density at non-spawning sites? Am I correct that only 10 fish were observed at SP1 and 3 fish at SP2 in all of 2018?
Line 278: Can you be more specific, what was the farthest distance?
Line 281-282: Does that mean that there were juvenile fish present within the group as well? That seems odd if it is truly a spawning aggregation? In lines 269-270 you state that reproductive behavior was observed in 12 instances, what is the difference between the 9 occasions stated here?
Line 286-287: Again, I am not sure what information the rate of tail flaps is supposed to be conveying or related to.
Line 300-303: Do you have any direct evidence of mottled groupers making either sound, on camera or during visual surveys? That would help support the connection. Otherwise, I think it would be helpful to acknowledge if there were any other fish in the area (any species) and explain why they are not candidates as the source of sound.
Line 342/Figure 8: you can remove the graphs for SP1 and SP2 since there is no data
Line 366: SP1 and SP2 don’t seem to show any sign that they are spawning sites for M. rubra, so I think that any discussion of those sites in comparison to MOL are not informative.
Line 451-452: With this data it is still uncertain if MOL is a spawning site or a pre-spawning aggregation, staging area or courtship arena (nemeth 2012), Spawning may still take place at a location well removed from MOL.
Round 2
Reviewer 4 Report
Thank you for your revisions and attention to detail. I believe the manuscript has been significantly improved. I only have two minor comments.
Line 261: “ranged from 10 minutes every 3 minutes to” This isn’t possible
Table 1: Total N. UVC transects with M. rubra at MOL should be 23 not 22